# Sex differences in chronic kidney disease awareness among US adults, 1999 to 2018

Sebastian Hödlmoser[1,2◦], Wolfgang C. Winkelmayer[3‡], Jarcy Zee[4‡], Roberto Pecoits-Filho[4,5‡], Ronald L. Pisoni[4‡], Friedrich K. Port[4‡], Bruce M. Robinson[4‡], Robin Ristl[6‡], Simon Krenn[2‡], Amelie Kurnikowski[2‡], Michał Lewandowski[2‡], Allison Ton[2,7‡], Juan Jesus Carrero[8‡], Eva S. Schernhammer[1,9,10]*, Manfred Hecking[2◦]

1 Department of Epidemiology, Center for Public Health, Medical University of Vienna, Vienna, Austria, 2 Department of Internal Medicine III, Clinical Division of Nephrology & Dialysis, Medical University of Vienna, Vienna, Austria, 3 Selzman Institute for Kidney Health, Section of Nephrology, Baylor College of Medicine, Houston, TX, United States of America, 4 Arbor Research Collaborative for Health, Ann Arbor, MI, United States of America, 5 School of Medicine, Pontificia Universidade Catolica do Parana, Curitiba, Brazil, 6 Center for Medical Statistics, Informatics and Intelligent Systems, Medical University of Vienna, Vienna, Austria, 7 Sydney School of Public Health, The University of Sydney, Sydney, Australia, 8 Department of Medical Epidemiology and Biostatistics, Karolinska Institutet, Stockholm, Sweden, 9 Department of Epidemiology, Harvard T.H. Chan School of Public Health, Boston, MA, United States of America, 10 Channing Division of Network Medicine, Brigham and Women's Hospital and Harvard Medical School, Boston, MA, United States of America

◦ These authors contributed equally to this work.
‡ These authors also contributed equally to this work.
* eva.schernhammer@meduniwien.ac.at

**Data Availability Statement:** The present study is based on NHANES data, which are publicly available at https://wwwn.cdc.gov/nchs/nhanes/Default.aspx. The code of our statistical analyses

## Abstract

### Background

Chronic kidney disease (CKD) is less prevalent among men than women, but more men than women initiate kidney replacement therapy. Differences in CKD awareness may contribute to this gender gap, which may further vary by race/ethnicity. We aimed to investigate trends in CKD awareness and the association between individual characteristics and CKD awareness among US men versus women.

### Methods and findings

We conducted a serial, cross-sectional analysis of 10 cycles (1999–2018) from the National Health and Nutrition Examination Survey (NHANES). Adult participants with CKD stages G3-G5 (estimated glomerular filtration rate [eGFR] <60 mL/min/1.73m$^2$) were included, unless they were on dialysis or medical information was missing. Serum creatinine was measured during NHANES medical exams. CKD stage was classified by eGFR, based on the CKD-EPI formula. CKD awareness was assessed with the question: "Have you ever been told by a health care professional you had weak or failing kidneys", asked in standardized NHANES questionnaires on each survey. Using logistic regression models, we evaluated the association between sex and CKD awareness, adjusting for potential confounders including age, race/ethnicity and comorbidities. We stratified CKD awareness by 5 predefined calendar-year periods and conducted all analyses for the complete study population as well as the Caucasian and African American subpopulations. We found that among

has been uploaded to the Protocols.io website and can be accessed using the following link: https://protocols.io/file/cp9hbg8yp.r.

**Funding:** This study was supported by a grant from the Austrian Science Fund https://m.fwf.ac.at/en/ with grant No. KL754-B, received by MH. The funders had no role in study design, data collection and analysis, decision to publish, or preparation of the manuscript.

**Competing interests:** No authors have competing interests.

101871 US persons participating in NHANES, 4411 (2232 women) had CKD in stages G3-G5. These participants were, on average, 73±10 years old, 25.3% reported diabetes, 78.0% reported hypertension or had elevated blood pressure during medical examinations and 39.8% were obese (percentages were survey-weighted). CKD awareness was more prevalent among those with higher CKD stage, younger age, diabetes, hypertension and higher body mass index. CKD awareness was generally low (<22.5%), though it increased throughout the study period, remaining consistently higher among men compared to women, with a decreasing gender gap over time (adjusted odds ratio [men-to-women] for CKD awareness = 2.71 [1.31–5.64] in period 1; = 1.32 [0.82–2.12] in period 5). The sex difference in CKD awareness was smaller in African American participants, in whom CKD awareness was generally higher. Using serum creatinine rather than eGFR as the CKD-defining exposure, CKD awareness increased with rising serum creatinine, in a close to identical fashion among both sexes during 1999–2008, while during 2009–2018, CKD awareness among women increased earlier than among men (i.e. with lower serum creatinine levels).

## Conclusions

CKD awareness is lower among US women than men. The narrowing gap between the sexes in more recent years and the results on CKD awareness by serum creatinine indicate that health care professionals have previously been relying on serum creatinine to inform patients about their condition, but in more recent years have been using eGFR, which accounts for women's lower serum creatinine levels due to their lower muscle mass. Additional efforts should be made to increase CKD awareness among both sexes.

## Introduction

Chronic kidney disease (CKD) ranks among the top ten most common chronic diseases in the US, currently afflicting over 37 million Americans [1]. On a global scale, an almost twofold increase in CKD prevalence has occurred over the last two decades. The burden of CKD is growing above rates that would be expected based on demographic changes and population ageing [2,3].

CKD occurs in consequence of kidney damage and is usually characterized by a gradual loss of kidney function, ultimately leading to kidney failure requiring kidney replacement therapy, (most often by hemodialysis) [4], in those individuals who have not previously died [5]. Importantly, while US women have a higher cumulative lifetime risk than men of developing CKD [6], their likelihood of initiating and receiving kidney replacement therapy as prevalent dialysis patients is actually lower, in the US [7] and elsewhere [7–9]. The higher CKD lifetime risk of women has been confirmed on the population level, and for non-US countries. Specifically, Carrero et al. reported a higher frequency of CKD in stages G3 to G5 for women compared with men, living in 20 countries and 4 continents [10]. These findings may partly be explained by miscalculation of the estimated glomerular filtration rate (eGFR) among women [11] and the lack of age-specific eGFR thresholds to define CKD [12]. Nevertheless, underlying reasons for the sex and gender 'disparity' that more women than men are classified as having CKD, but roughly 60% men versus 40% women initiate [10] and also receive kidney replacement [7], are subject of debate among advocators of biology on one side and advocators of psycho-socio-economic factors on the other.

One of the earliest studies reporting that women in the US had a lower dialysis initiation rate than men overtly stated that this finding was a consequence of injustice and discrimination, specifically describing the calculation of a 'discrimination index' under the subheading 'Measures of Distribution and Injustice' in the Methods section [13]. This earlier report also identified a lower dialysis initiation rate in persons with kidney failure of older age and African American race [13], who have an even higher CKD risk than Caucasians [6,14,15]. Women have been shown to be less prepared regarding their vascular access than men at dialysis initiation [7,16], and to start dialysis with lower eGFR than men [7,17–19]. Besides initiating dialysis *later* than men, more women than men might also be remaining with low eGFR (i.e. <15 mL/min/1.73m$^2$), without initiating dialysis, among those who were previously referred to a nephrologist.

The pitfalls in correctly determining eGFR from serum creatinine [12,20,21], specifically an overestimation of poor kidney function among women [11] would lead to more women than men being prepared for and initiating dialysis. As the opposite is the case, it has been postulated that CKD is unrecognized in women, because healthcare professionals may subjectively estimate kidney function from serum creatinine levels, which are higher in men due to their higher muscle mass, rather than considering estimated glomerular filtration rates.

Prior research documented generally low awareness rates of CKD in the US population with little improvement over time [22,23], even in the most recent analysis through the year 2016 [24]. Specifically, CKD awareness rates consistently amounted to 10% and below for patients with CKD stage G3, and even lower for earlier stage CKD where level of albuminuria constitutes the disease-qualifying criterion. Whether consistent sex differences exist, remains less well understood [23]. In the present study using population-representative data from a large national health assessment program, we aimed at examining trends in CKD awareness from 1999–2018 and the association between CKD awareness and individual characteristics, such as socioeconomic and lifestyle risk factors, among US men *versus* women, overall, and by race/ethnicity.

## Methods

The National Health and Nutrition Examination Survey (NHANES) is a serial, cross sectional, representative study of non-institutionalized US persons. Currently, NHANES data are available from 1999 to 2018 [25].

### Study population for analysis

For the present study, we limited the analysis to NHANES participants aged 20 or older and to those who had been seen in mobile examination centers to have their laboratory measurements taken, including serum creatinine, which was used to calculate eGFR (see below). CKD stages 1 and 2 were defined as eGFR >60 mL/min/1.73m$^2$ in conjunction with persistent albuminuria and/or kidney damage. As the single-time measurements of albuminuria in NHANES are prone to misclassification of albuminuria status, we restricted our study to CKD stages G3 to G5, based solely on eGFR. Participants who answered 'Yes' to the question 'In the past 12 months, have you received dialysis (either hemodialysis or peritoneal dialysis)?' were excluded (N = 118), leaving 4411 subjects for descriptive statistics. N = 191 subjects had no information on body mass index (BMI) and were excluded from logistic regression, leaving 4220 participants across all NHANES cycles (N = 3840, N = 317 and N = 63 cases with CDK stage G3, G4 and G5, respectively) for our final analyses (= study sample, see **Fig 1**).

## CKD staging

For CKD staging, the CKD-EPI formula [26] was used to calculate eGFR (in mL/min/1.73m$^2$) for each subject, based on serum creatinine laboratory measurements (SCR, in mg/dL). In order to standardize towards a shared measurement referent across years, SCR was corrected for NHANES cycles 1999–2000, 2005–2006 and 2017–2018 according to the NHANES analytical guidelines as follows [27–29].

- 1999–2000: $SCR_{corr} = 1.013 * SCR_{orig} + 0.147$

- 2005–2006: $SCR_{corr} = 0.978 * SCR_{orig}—0.016$

- 2017–2018: $SCR_{corr} = 1.051 * SCR_{orig}—0.06945$

CKD stage G3 was defined as eGFR <60 and ≥30 mL/min/1.73m$^2$, CKD stage G4 as eGFR <30 and ≥15 mL/min/1.73m$^2$ and CKD stage G5 as eGFR <15 mL/min/1.73m$^2$. Due to data limitations, for subjects with 80 or 85 years and above, age was set to 80/85 in the eGFR formula, depending on data cycle.

## CKD awareness

CKD awareness was based on the question 'Have you ever been told by a doctor or other health professional that you had weak or failing kidneys? (Do not include kidney stones, bladder infections, or incontinence.)'. Subjects who answered 'Yes' and were classified in one of the CKD stages G3—G5 based on eGFR were defined as being aware of their CKD (= CKD aware).

## Covariates

We defined age groups as [20 to 49 years], [50 to 64 years], [65 to 79 years] and 80+ years. Participants were considered hypertensive if they were ever told by a physician to have high blood

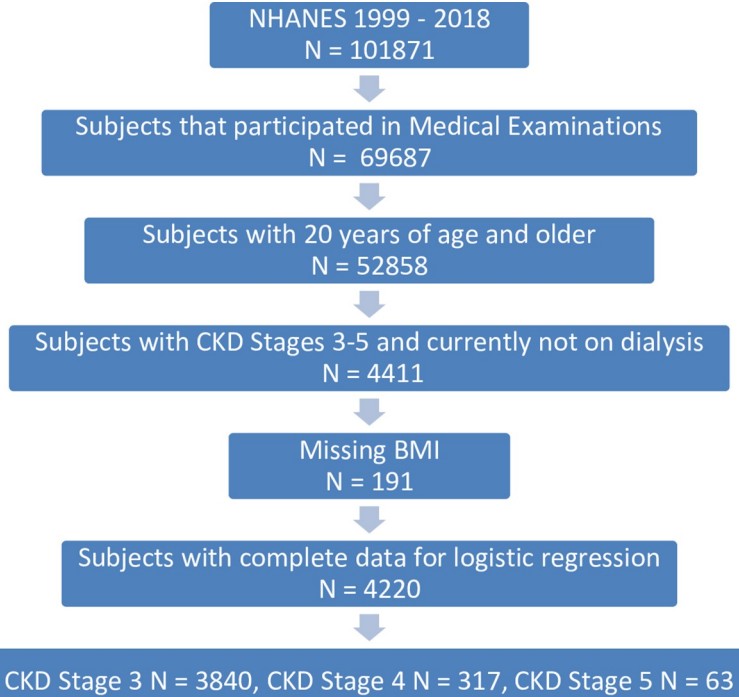

**Fig 1. Flow chart of the study sample.**

pressure, or if in the medical examination their mean blood pressure (of up to four measurements) was >140 mm Hg in systole or >90 mm Hg in diastole. Participants were classified as having diabetes, per the NHANES definition, if they were self reportedly told so by a doctor. Participants who answered 'Every Day' or 'Some Days' to the question 'Do you now smoke cigarettes?' were considered smokers (Yes/No).

## Statistical methods

To increase sample size and simplify trend analysis and tables, we aggregated the data in 4-,6-, 8-, 10- and 20-year periods and rescaled the respective NHANES examinations sample weights [30]. To examine participant characteristics, we determined the crude number of subjects in the dataset together with survey weighted percentages (**Tables 1** and **2**). The weighted percentages are the (representative) fraction of CKD aware US persons, respective to the entire CKD stage G3—G5 population among US persons in general. To examine differences in sex-specific awareness in Caucasian and African American subpopulations, we determined the number of CKD aware participants and weighted awareness proportions in those groups (**Table 2**). Due to low sample sizes of the other races/ethnicities, we limited the subpopulation analysis to Caucasians and African Americans. We used multivariable weighted logistic regression to estimate odds ratios (ORs) and corresponding 95% confidence intervals (CIs) of CKD awareness by sex, adjusted for 4-year calendar year interval, age, CKD stage and the risk factors race/ethnicity, BMI, diabetes and hypertension. Specifically, we calculated six logistic regression models, where in each model we recoded the exposures such that men and women could be compared within a second variable (period, stage, age, diabetes, hypertension, BMI). Within each variable, women in one level of the variable represented the baseline. Odds ratios were adjusted for all other characteristics. We calculated all models for our NHANES study sample with a race/ethnicity factor variable and for Caucasians and African Americans separately (**Fig 2** and supporting information **S2 Table**). Moreover we compared survey-weighted CKD awareness by year with fully adjusted male-to-female awareness ORs and 95% CIs by year (**Fig 3**). Again, we calculated all models for our NHANES study sample, and separately for the Caucasian and African American subpopulations. Further we compared survey-weighted CKD awareness of men and women by serum creatinine intervals in the first in second half of the observation period (**Fig 4**). All calculations were done using R 4.0.3 [31]. For survey weighted analysis we used the *srvyr* and *survey* R-libraries [32]. P-values less than 0.05 were considered significant.

## Results

### Characteristics of the study sample

Among 101871 surveyed US persons, 2232 women and 2179 men were above 20 years and classified as having CKD stages G3 to G5 (**Fig 1**). **Table 1** shows the characteristics of these study participants (**left column**) and their distribution over 4-year periods (**upper row**). As demonstrated in the **left column of Table 1**, most of the study participants with CKD had CKD stage G3 (92.7%), more than half were Caucasian (80.8%) and the largest age group was between 65 and 79 years old, representing 44.5% of the study sample. Diabetes was reported by 25.3%, while 78.0% reported to have hypertension or had elevated blood pressure during medical examinations (all percentages are weighted, as indicated in the Methods). Mean BMI was $29.4\pm6.5$ kg/m$^2$, with three quarters being overweight (35.8%) or obese (39.8%). Throughout all periods of analysis, the overall proportion of women having CKD in stages G3 to G5 was slightly higher than for men, as has previously been reported for this dataset [33]. The comparison of the study sample with the remaining NHANES population that underwent medical examinations is provided in supporting information **S1 Table** and shows that the remaining

**Table 1. Characteristics of NHANES participants with CKD in stages G3 –G5 (study sample) and of those individuals among the study sample who were CKD aware, over time and by sex.**

| | Total | 1999–2002 | | 2003–2006 | | 2007–2010 | | 2011–2014 | | 2015–2018 | |
|---|---|---|---|---|---|---|---|---|---|---|---|
| | | Men | Women | Men | Women | Men | Women | Men | Women | Men | Women |
| | N (weighted %) | N = 393 | N = 442 | N = 418 | N = 477 | N = 440 | N = 532 | N = 423 | N = 451 | N = 412 | N = 423 |
| | | CKD aware men / women: N (weighted %) within each variable and within each sex | | | | | | | | | |
| **Total** | 4411 (100.0) | 70 (17.6) | 40 (8.5) | 53 (13.6) | 40 (9.2) | 67 (14.2) | 57 (9.0) | 68 (14.0) | 68 (11.7) | 103 (22.5) | 92 (18.0) |
| **CKD Stage** | | | | | | | | | | | |
| G3 | 4007 (92.7) | 43 (13.2) | 21 (6.0) | 37 (11.0) | 29 (7.1) | 51 (11.1) | 30 (5.3) | 49 (11.3) | 47 (9.1) | 85 (19.8) | 68 (15.7) |
| G4 | 338 (6.2) | 16 (41.3) | 8 (32.6) | 14 (41.3) | 9 (30.2) | 15 (65.3) | 23 (42.8) | 12 (47.2) | 17 (42.1) | 17 (73.0) | 23 (47.9) |
| G5 | 66 (1.1) | 11 (97.0) | 11 (72.1) | 2 (93.1) | 2 (81.7) | 1 (49.6) | 4 (90.5) | 7 (83.2) | 4 (69.2) | 1 (33.9) | 1 (25.3) |
| **Race/Ethnicity** | | | | | | | | | | | |
| Caucasian | 2762 (80.8) | 32 (15.3) | 18 (6.2) | 33 (12.3) | 25 (8.8) | 40 (13.9) | 25 (7.3) | 32 (12.3) | 30 (9.5) | 47 (20.8) | 41 (16.3) |
| African American | 827 (9.3) | 18 (28.2) | 14 (21.8) | 15 (30.6) | 9 (13.9) | 12 (14.8) | 15 (15.2) | 18 (28.4) | 24 (26.7) | 26 (30.2) | 26 (27.5) |
| Mexican American | 378 (2.8) | 16 (45.2) | 4 (10.3) | 3 (17.2) | 4 (8.7) | 9 (22.0) | 9 (24.2) | 6 (25.0) | 7 (30.1) | 12 (36.0) | 12 (24.7) |
| Other | 444 (7.1) | 4 (25.2) | 4 (15.9) | 2 (7.6) | 2 (9.5) | 6 (12.7) | 8 (17.1) | 12 (11.3) | 7 (10.5) | 18 (22.2) | 13 (18.1) |
| **Age Mean (SD)** | 73.0 (10.2) | | | | | | | | | | |
| [20,49] | 142 (4.6) | 9 (49.4) | 2 (9.5) | 3 (13.2) | 1 (7.8) | 1 (15.4) | 5 (44.2) | 4 (15.1) | 8 (21.8) | 9 (54.1) | 5 (43.9) |
| [50,64] | 646 (18.7) | 11 (18.3) | 12 (17.8) | 11 (20.7) | 11 (17.8) | 15 (14.2) | 10 (6.4) | 16 (11.6) | 17 (14.0) | 20 (27.2) | 16 (20.4) |
| [65,79] | 1932 (44.5) | 35 (16.5) | 15 (6.9) | 23 (14.2) | 16 (9.2) | 34 (16.2) | 28 (9.9) | 36 (17.6) | 26 (10.6) | 47 (19.7) | 38 (14.2) |
| 80+ | 1691 (32.1) | 15 (9.9) | 11 (5.7) | 16 (8.6) | 12 (5.3) | 17 (10.9) | 14 (6.4) | 12 (9.9) | 17 (9.7) | 27 (17.0) | 33 (20.2) |
| **Diabetes** | | | | | | | | | | | |
| No | 3142 (74.7) | 39 (13.6) | 26 (6.9) | 29 (8.4) | 22 (7.7) | 33 (10.2) | 28 (6.1) | 34 (11.1) | 33 (8.0) | 56 (19.1) | 45 (13.1) |
| Yes | 1269 (25.3) | 31 (37.9) | 14 (16.0) | 24 (27.8) | 18 (15.4) | 34 (25.4) | 29 (17.5) | 34 (20.5) | 35 (22.3) | 47 (28.3) | 47 (32.3) |
| **Hypertension** | | | | | | | | | | | |
| No | 868 (22.0) | 7 (8.8) | 5 (4.0) | 6 (8.6) | 4 (5.4) | 10 (8.7) | 3 (2.5) | 3 (2.6) | 9 (10.7) | 13 (17.0) | 7 (7.9) |
| Yes | 3543 (78.0) | 63 (20.8) | 35 (9.7) | 47 (15.5) | 36 (10.1) | 57 (16.4) | 54 (10.3) | 65 (17.4) | 59 (12.0) | 90 (24.3) | 85 (20.2) |
| **BMI Missing** | 191 | | | | | | | | | | |
| Mean (SD) | 29.4 (6.5) | | | | | | | | | | |
| <25 | 1048 (24.4) | 8 (5.4) | 3 (1.2) | 7 (10.8) | 9 (7.6) | 14 (13.6) | 12 (5.9) | 13 (13.8) | 6 (3.9) | 20 (17.9) | 11 (10.8) |
| Overweight (25,30] | 1523 (35.8) | 29 (19.9) | 16 (9.8) | 20 (12.3) | 15 (7.8) | 20 (10.4) | 14 (5.1) | 20 (13.5) | 16 (12.1) | 29 (17.3) | 22 (12.7) |
| Obese >30 | 1649 (39.8) | 24 (23.7) | 15 (10.3) | 19 (14.7) | 15 (12.0) | 27 (16.9) | 31 (14.5) | 34 (15.9) | 43 (16.0) | 51 (28.3) | 55 (23.5) |
| **Smoker Missing** | 2227 | | | | | | | | | | |
| No | 1801 (82.2) | 32 (14.3) | 15 (12.8) | 26 (13.8) | 12 (8.3) | 47 (17.6) | 18 (11.2) | 36 (17.2) | 23 (10.6) | 57 (25.3) | 34 (18.4) |
| Yes | 383 (17.8) | 11 (36.3) | 4 (9.3) | 8 (29.7) | 3 (9.0) | 8 (26.0) | 6 (6.6) | 4 (10.9) | 6 (19.5) | 10 (17.1) | 6 (21.5) |
| **Income Missing** | 169 | | | | | | | | | | |
| >20.000 | 2920 (75.2) | 42 (16.8) | 25 (12.0) | 36 (13.1) | 21 (8.7) | 48 (13.2) | 33 (8.0) | 57 (14.5) | 43 (11.7) | 68 (20.4) | 56 (15.5) |
| <20.000 | 1322 (24.8) | 23 (21.9) | 11 (5.5) | 17 (17.8) | 16 (10.1) | 19 (20.4) | 23 (11.8) | 11 (12.3) | 24 (11.6) | 28 (34.3) | 35 (27.0) |
| **Education** | | | | | | | | | | | |
| High School or higher | 2908 (74.6) | 35 (18.2) | 23 (8.2) | 36 (12.3) | 25 (10.1) | 32 (10.5) | 34 (8.4) | 46 (13.1) | 48 (11.8) | 77 (23.1) | 65 (16.3) |
| less than High School | 1503 (25.4) | 35 (16.5) | 17 (9.0) | 17 (17.4) | 15 (7.4) | 35 (25.2) | 23 (10.4) | 22 (17.1) | 20 (11.5) | 26 (19.9) | 27 (27.4) |
| **Health Insurance** | | | | | | | | | | | |
| No | 201 (4.3) | 4 (32.3) | NA (0.0) | 2 (17.3) | 3 (15.1) | 2 (6.7) | 4 (15.9) | 7 (28.1) | 5 (16.0) | 6 (42.2) | 4 (17.4) |
| Yes | 4210 (95.7) | 66 (17.0) | 40 (9.1) | 51 (13.5) | 37 (9.0) | 65 (14.5) | 53 (8.6) | 61 (13.4) | 63 (11.5) | 97 (21.1) | 88 (18.0) |
| **Healthcare Visits Missing** | 14 | | | | | | | | | | |
| 0 | 140 (2.9) | 1 (8.2) | 1 (4.1) | 2 (7.7) | NA (0.0) | NA (0.0) | NA (0.0) | 2 (10.1) | NA (0.0) | 2 (31.9) | 1 (5.9) |
| 1–3 | 1420 (33.5) | 11 (7.3) | 10 (7.0) | 11 (6.8) | 9 (8.1) | 13 (9.4) | 9 (6.7) | 13 (9.0) | 11 (5.2) | 28 (20.3) | 21 (9.6) |
| 4–12 | 1713 (39.2) | 31 (21.9) | 15 (10.9) | 14 (11.6) | 16 (7.4) | 29 (15.5) | 17 (5.6) | 22 (12.0) | 35 (14.8) | 42 (20.1) | 44 (24.9) |
| >12 | 1124 (24.4) | 27 (28.9) | 14 (7.8) | 26 (24.0) | 15 (15.7) | 25 (19.9) | 31 (17.8) | 31 (24.7) | 22 (15.3) | 31 (28.9) | 24 (19.8) |

*(Continued)*

**Table 1.** (Continued)

|  | Total | 1999–2002 | 2003–2006 | 2007–2010 | 2011–2014 | 2015–2018 |
|---|---|---|---|---|---|---|
| **Overall CKD Awareness (weighted %)** |  | 11.2% | 10.7% | 10.8% | 12.8% | 19.8% |

Entire Column 1 (left) and row 3: NHANES participants with CKD in stages G3 –G5. Columns 2–11 (right), Rows 5–47: *CKD aware* individuals among the study sample participants (N's and survey-weighted percentages), by 4-year study period and sex. Row 48: Overall CKD awareness over time.

NHANES participants were younger, less frequently had diabetes or hypertension and had lower BMI. While African Americans were equally represented in the study sample compared to the remaining NHANES population, they were decidedly overrepresented in the groups of more advanced CKD stages (supporting infromation **S2 Table**), in agreement with a previous report [6].

### Descriptive analysis of CKD awareness, by sex and race/ethnicity

The other rows and columns of **Table 1** report on *CKD aware* men and women, in absolute unweighted counts as well as weighted percentages. Overall awareness increased from 11.2% in 1999–2002 to 19.8% in 2015–2018. The most substantial rise in awareness took place in the last 4-year period (last row of **Table 1**). Awareness increased with higher CKD stage and was generally higher among African Americans, younger age groups, and persons with diabetes, hypertension, and those with higher BMI. Socioeconomic factors did not show a clear pattern with respect to CKD awareness over time and by sex, apart from the overall increase. Throughout all 4-year periods CKD awareness among men was higher than among women, although the male to female difference of 9.1 percentage points in the first 4-year period decreased to 4.5 percentage points in the most recent 4-year period. Women more than doubled their CKD awareness over the two decades of analysis, which was mostly due to their substantial growth in CKD awareness during the last 4-year period. CKD awareness in CKD stage 3 was below 20% and especially low among women, although it almost tripled from 6.0% in 1999–2002 to 15.7% in 2015–2018. Awareness in CKD stage G4 fluctuated among men but overall increased from 41.3% in the first 4-year period to 73.0% in the last 4-year period, as did awareness among women, which increased from 32.6% in the first 4-year period to 47.9% in the last 4-year period.

Analysis by race/ethnicity revealed that the most obvious sex difference in CKD awareness was observed among Caucasians, while CKD awareness was more evenly distributed between the sexes among African Americans, Mexican Americans and other races/ethnicities throughout the period of analysis (**Table 1**). **Table 2** shows awareness rates among Caucasian and African American subpopulations. The trend towards increased male awareness, as already indicated in **Table 1**, was less obvious among African Americans compared to Caucasians. In contrast to African Americans, almost all Caucasians with CKD stage G5 were aware of their CKD.

### Regression model analysis of CKD awareness, by sex and race/ethnicity

To disentangle the independent associations of the exposures on each other, we modeled CKD awareness via weighted logistic regression. As socioeconomic factors did not appear to have a strong influence in the univariate analysis, we chose parsimonious models which, besides sex and calendar time interval, incorporated factor variables for CKD stage, age group and the risk factors race/ethnicity, diabetes, hypertension, and BMI. The estimates are shown in **Fig 2A** and the corresponding numeric ORs and 95% CIs are reported in **S3 Table**. Briefly, this analysis confirmed the associations from **Table 1**, as the odds for CKD awareness increased with

**Table 2. Characteristics of CKD aware Caucasians (left) and African Americans (right), by 8- or 6-year study period and sex.**

| | 1999–2006 | | 2007–2012 | | | | 1999–2006 | | 2007–2012 | | 2013–2018 | |
|---|---|---|---|---|---|---|---|---|---|---|---|---|
| | Caucasian | | | | | | African American | | | | | |
| | Men | Women | Men | Women | Men | Women | Men | Women | Men | Women | Men | Women |
| | N = 573 | N = 618 | N = 391 | N = 455 | N = 347 | N = 378 | N = 134 | N = 149 | N = 122 | N = 162 | N = 124 | N = 136 |
| | CKD aware men / women, N (weighted %) within each variable and each sex | | | | | | | | | | | |
| **Total** | 65 (13.7) | 43 (7.5) | 55 (12.8) | 39 (8.9) | 64 (18.0) | 57 (12.8) | 33 (29.3) | 23 (17.9) | 19 (16.0) | 28 (18.8) | 37 (33.0) | 37 (27.7) |
| **CKD Stage** | | | | | | | | | | | | |
| G3 | 52 (11.4) | 30 (5.9) | 38 (9.7) | 26 (7.1) | 51 (15.7) | 48 (11.4) | 19 (23.1) | 15 (14.0) | 13 (12.7) | 16 (11.0) | 33 (32.6) | 23 (21.0) |
| G4 | 10 (33.0) | 8 (22.9) | 15 (53.7) | 12 (32.6) | 11 (68.4) | 8 (38.3) | 9 (48.5) | 3 (26.0) | 4 (42.9) | 10 (71.9) | 3 (37.6) | 12 (53.0) |
| G5 | 3 (100.0) | 5 (73.1) | 2 (100.0) | 1 (100.0) | 2 (100.0) | 1 (100.0) | 5 (87.9) | 5 (100.0) | 2 (49.1) | 2 (56.2) | 1 (32.5) | 2 (68.7) |
| **Age** | | | | | | | | | | | | |
| [20,49] | 3 (38.5) | 1 (7.2) | 1 (15.9) | 2 (25.2) | 2 (42.1) | 3 (21.8) | 5 (35.2) | 1 (26.3) | NA (0.0) | 4 (66.7) | 3 (57.4) | 4 (68.1) |
| [50,64] | 8 (14.8) | 9 (15.1) | 7 (10.7) | 6 (11.4) | 6 (15.7) | 9 (11.8) | 9 (56.1) | 10 (39.2) | 7 (25.7) | 8 (18.2) | 14 (42.4) | 8 (19.3) |
| [65,79] | 29 (14.3) | 15 (7.1) | 28 (14.6) | 17 (7.8) | 36 (20.6) | 19 (11.1) | 17 (21.8) | 11 (14.4) | 11 (16.2) | 10 (11.3) | 14 (24.0) | 16 (25.1) |
| 80+ | 25 (9.1) | 18 (4.7) | 19 (10.9) | 14 (7.7) | 20 (12.5) | 26 (14.5) | 2 (10.9) | 1 (2.8) | 1 (4.8) | 6 (23.0) | 6 (24.7) | 9 (31.6) |
| **Diabetes** | | | | | | | | | | | | |
| No | 42 (9.4) | 31 (6.4) | 32 (10.6) | 21 (6.1) | 34 (13.6) | 35 (9.9) | 17 (26.6) | 10 (14.1) | 8 (11.6) | 9 (11.6) | 23 (31.0) | 17 (21.6) |
| Yes | 23 (31.0) | 12 (13.7) | 23 (19.3) | 18 (18.8) | 30 (26.5) | 22 (23.5) | 16 (34.9) | 13 (24.6) | 11 (22.3) | 19 (25.5) | 14 (36.7) | 20 (37.0) |
| **Hypertension** | | | | | | | | | | | | |
| No | 11 (8.6) | 7 (4.8) | 8 (7.6) | 2 (5.5) | 8 (10.7) | 7 (6.4) | NA (0.0) | NA (0.0) | NA (0.0) | 1 (16.4) | 3 (20.0) | 2 (17.9) |
| Yes | 54 (15.8) | 36 (8.2) | 47 (14.6) | 37 (9.9) | 56 (20.7) | 50 (14.5) | 33 (32.6) | 23 (19.3) | 19 (18.5) | 27 (18.9) | 34 (35.3) | 35 (28.7) |
| **BMI** | | | | | | | | | | | | |
| <25 | 8 (8.0) | 7 (3.9) | 15 (17.4) | 5 (2.7) | 10 (11.4) | 9 (7.9) | 1 (3.1) | 3 (8.6) | 3 (8.3) | 3 (12.0) | 7 (23.9) | 3 (18.1) |
| Overweight—(25,30] | 27 (13.6) | 16 (7.0) | 17 (10.0) | 9 (7.7) | 20 (15.7) | 11 (7.9) | 16 (38.0) | 11 (33.5) | 5 (12.0) | 7 (18.8) | 10 (26.6) | 12 (46.8) |
| Obese—>30 | 23 (16.6) | 16 (10.3) | 19 (13.3) | 24 (15.0) | 33 (23.2) | 35 (18.1) | 11 (29.5) | 8 (12.5) | 11 (23.8) | 17 (20.4) | 19 (39.9) | 22 (24.8) |
| **Smoker** | | | | | | | | | | | | |
| No | 36 (12.4) | 14 (8.4) | 40 (16.0) | 14 (9.2) | 44 (22.8) | 23 (13.4) | 13 (32.4) | 7 (14.3) | 11 (20.4) | 9 (15.4) | 14 (33.9) | 14 (33.6) |
| Yes | 11 (33.4) | 5 (9.4) | 5 (31.1) | 4 (14.5) | 3 (6.8) | 2 (6.9) | 4 (26.9) | 1 (6.7) | 4 (20.3) | 4 (21.9) | 7 (29.2) | 3 (25.0) |
| **Income** | | | | | | | | | | | | |
| >20.000 | 43 (13.2) | 27 (9.2) | 42 (11.6) | 27 (9.1) | 50 (17.2) | 37 (11.1) | 24 (28.8) | 12 (19.4) | 13 (15.4) | 16 (18.9) | 24 (35.0) | 21 (25.3) |
| <20.000 | 20 (17.3) | 13 (5.4) | 13 (23.4) | 12 (8.4) | 13 (23.4) | 20 (19.4) | 8 (34.8) | 9 (16.4) | 6 (17.5) | 12 (18.8) | 11 (28.4) | 15 (34.1) |
| **Education** | | | | | | | | | | | | |
| High School or higher | 47 (13.9) | 32 (8.5) | 27 (9.4) | 26 (8.8) | 53 (18.7) | 48 (12.0) | 18 (31.0) | 11 (23.4) | 10 (14.1) | 20 (24.1) | 30 (36.7) | 28 (27.4) |
| less than High School | 18 (13.2) | 11 (5.3) | 28 (24.7) | 13 (9.3) | 11 (13.7) | 9 (19.2) | 15 (27.6) | 12 (14.1) | 9 (19.0) | 8 (12.3) | 7 (22.3) | 9 (28.6) |
| **Health Insurance** | | | | | | | | | | | | |
| No | 2 (28.3) | 1 (4.6) | NA (0.0) | 2 (13.2) | 3 (44.9) | 3 (20.7) | 2 (43.4) | 2 (30.4) | 3 (28.7) | 1 (8.3) | 3 (35.7) | 2 (19.8) |
| Yes | 63 (13.4) | 42 (7.6) | 55 (13.0) | 37 (8.8) | 61 (16.7) | 54 (12.6) | 31 (28.7) | 21 (16.9) | 16 (14.6) | 27 (19.5) | 34 (32.6) | 35 (28.3) |
| **Healthcare Visits** | | | | | | | | | | | | |
| 0 | 3 (10.0) | 1 (2.5) | 1 (5.8) | NA (0.0) | 2 (39.7) | NA (0.0) | NA (0.0) | NA (0.0) | NA (0.0) | NA (0.0) | NA (0.0) | 1 (21.0) |
| 1–3 | 10 (4.5) | 13 (7.5) | 10 (7.5) | 8 (6.0) | 15 (16.8) | 11 (6.3) | 9 (24.9) | 4 (14.2) | 1 (3.0) | 3 (6.9) | 10 (24.5) | 8 (17.4) |
| 4–12 | 25 (15.8) | 18 (7.0) | 22 (12.5) | 15 (9.4) | 26 (14.4) | 31 (17.4) | 11 (28.9) | 9 (19.1) | 9 (20.1) | 15 (21.1) | 15 (36.3) | 18 (33.8) |
| >12 | 27 (23.7) | 11 (9.3) | 22 (20.1) | 16 (12.6) | 21 (23.4) | 15 (15.0) | 13 (40.3) | 10 (21.6) | 9 (30.4) | 10 (31.2) | 12 (47.4) | 10 (33.7) |

higher CKD stage, younger age, diabetes, hypertension and higher BMI. The analysis further indicated that men had higher odds for CKD awareness than women and that the difference in CKD awareness between men and women diminished over time.

Subgroup analyses by racial/ethnic group are shown in Fig 2B (Caucasians) and Fig 2C (African Americans) and indicated that sex differences in CKD awareness were generally

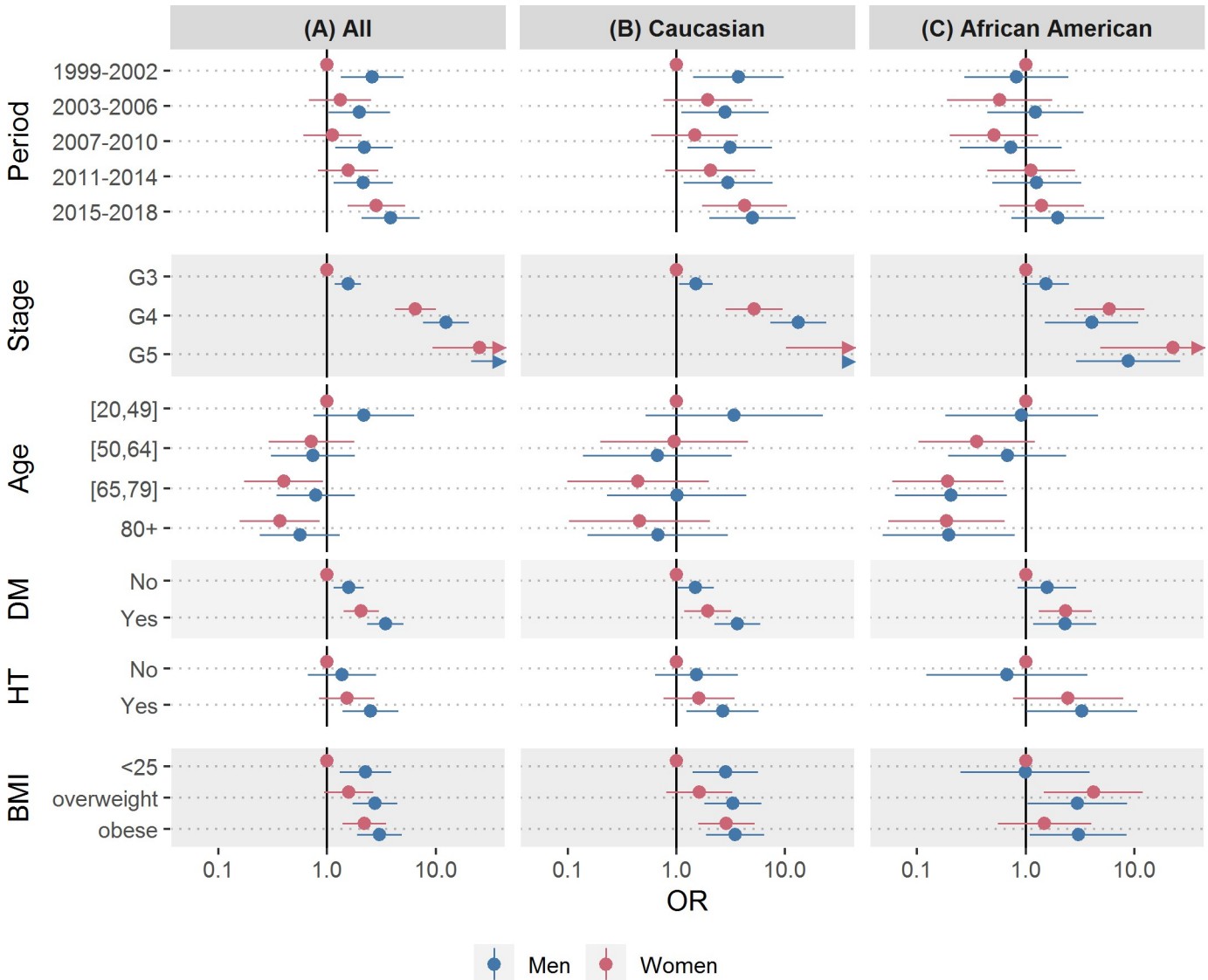

**Fig 2. Association of chronic kidney disease awareness and patient characteristics.** Association of chronic kidney disease (CKD) awareness by sex with period, CKD stage, age group, Diabetes Mellitus (DM), Hypertension (HT) and BMI in all NHANES participants (A), Caucasians (B) and African Americans (C), 1999–2018; Odds ratios were adjusted for all other characteristics shown, odds ratios for all participants (A) were further adjusted for race/ethnicity. The numerical values for these odds ratios and 95% confidence intervals are shown in supporting information S3 Table.

smaller among African Americans, compared with Caucasians. The overall increase in CKD awareness over time appeared to be driven by an increase in CKD awareness among Caucasian women (female awareness in period 5 vs. period 1: OR = 2.84 [1.55–5.22] among all participants, OR = 4.25 [1.72–10.48] among Caucasians, OR = 1.45 [0.57–3.44] among African Americans).

## Summary analysis of time trends in CKD awareness, by sex and race/ethnicity

To provide a comprehensive summary of male versus female CKD awareness over time, we calculated weighted proportions of CKD aware NHANES participants (with 95% CIs) by sex

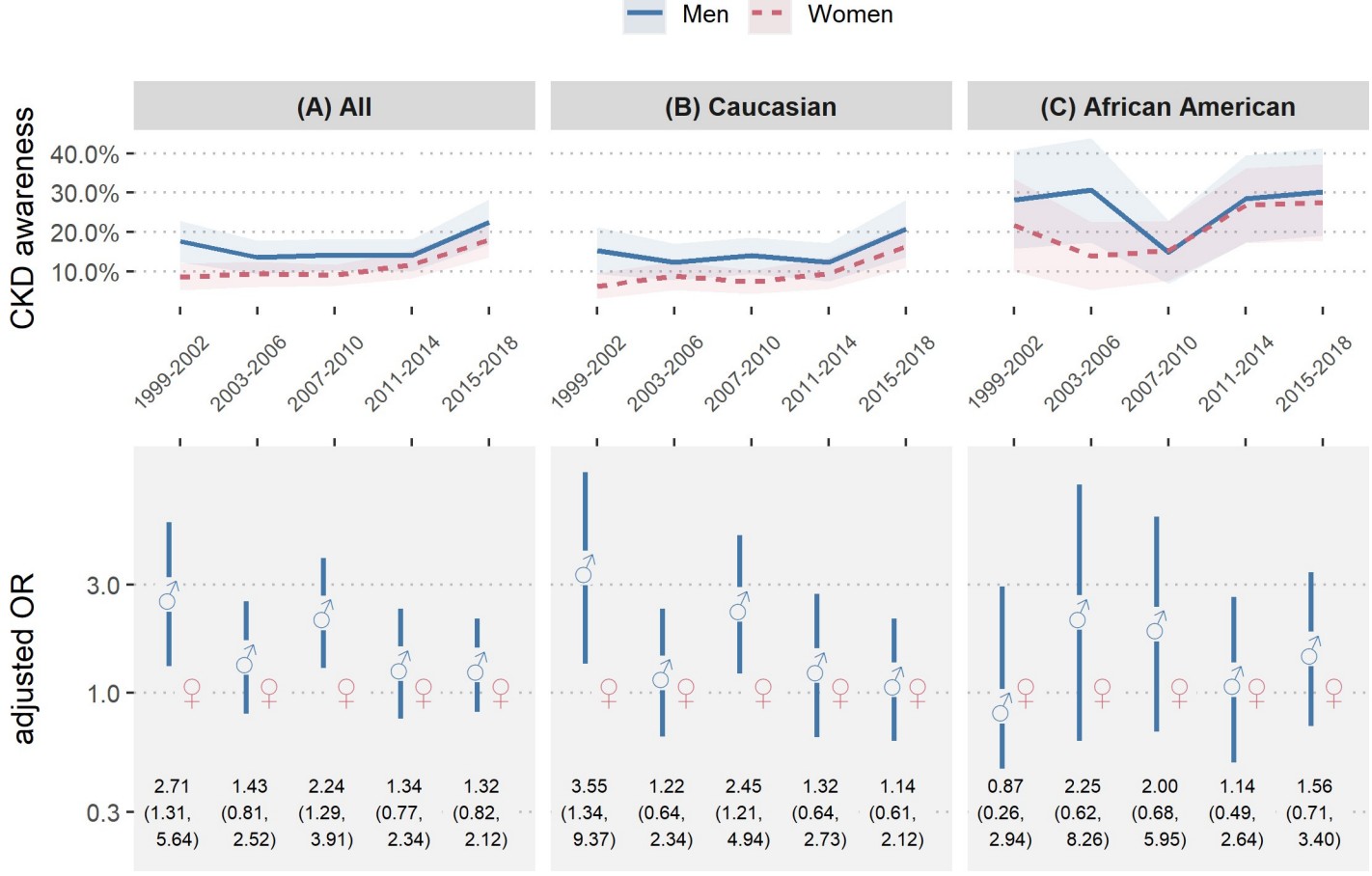

**Fig 3. Chronic kidney disease awareness of men versus women over time.** Chronic kidney disease (CKD) awareness of men versus women in NHANES 1999–2018 of all study participants (A), Caucasians (B) and African Americans (C); (top) Unadjusted CKD awareness over time with 95% confidence interval (bottom) CKD awareness odds ratios of men versus women by year with 95% confidence interval; Odds ratios were adjusted for CKD disease stage, age, diabetes, hypertension and BMI; Odds ratios for all study participants were further adjusted for race/ethnicity.

and 4-year-period, from 1999 to 2018 (**Fig 3, upper row**), in conjunction with fully adjusted odds ratios for male-to-female awareness (**Fig 3, lower row**). Subgroup analyses by racial/ethnic origin are shown in **Fig 3B** (Caucasians) and **Fig 3C** (African Americans). In the 4-year periods 1 (1999–2002) and 3 (2007–2010), men were significantly more CKD aware than women (male-to-female OR = 2.71 [1.31–5.64] in period 1 and OR = 2.24 [1.29–3.91] in period 3). However, the sex-specific differences in CKD awareness were not significant thereafter. The fully adjusted model once more indicated (compare **Fig 2**), that the sex-specific difference in CKD awareness among Caucasians might be the largest driver of the overall difference, as the observed significant ORs in the full study sample persisted within Caucasians (male-to-female OR = 3.55 [1.34–9.37] in period 1 and OR = 2.45 [1.21–4.94] in period 3) but not in African Americans. However, this hypothesis could not formally be confirmed, because testing sex and race/ethnicity interactions did not show significant effect modification.

## CKD awareness by serum creatinine range

A possible explanation for differing CKD awareness between the sexes is the use of serum creatinine instead of eGFR to assess kidney function, as was the case before standardized eGFR

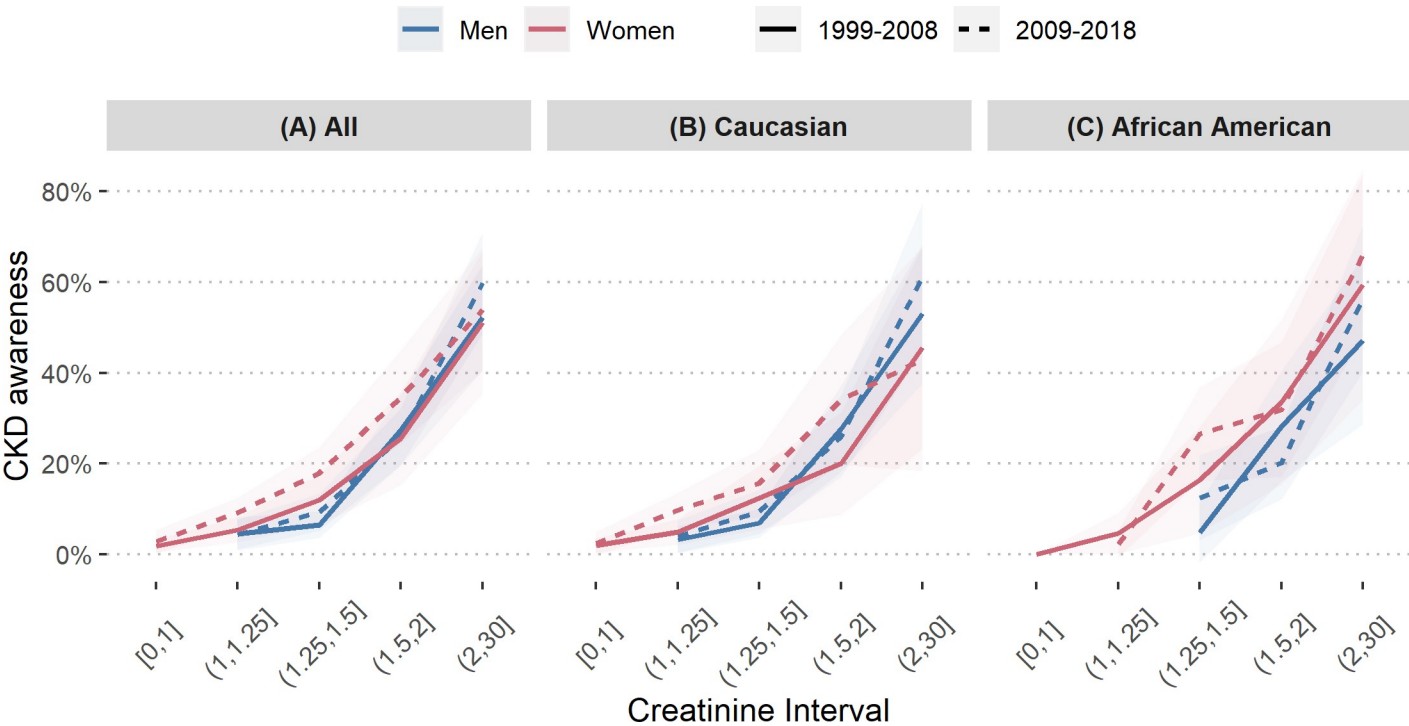

**Fig 4. Chronic kidney disease awareness of men versus women by creatinine.** Chronic kidney disease awareness of men versus women in NHANES 1999–2008 and 2009–2018, by creatinine interval, of all study participants (A), Caucasians (B) and African Americans (C).

formulas became implemented into routine clinical practice. **Fig 4** shows CKD awareness of men and women by serum creatinine interval in the first and second half of the observation period. From 1999–2008 the awareness of men and women overlapped to a great extent, while in 2009–2018 female awareness increased at earlier levels of serum creatinine. Further, the significant sex-differences from the models in **Fig 3** disappeared when they were adjusted for serum creatinine instead of CKD stage.

## Discussion

In the present study, we used nationally representative data to conduct a comprehensive assessment of the difference in CKD awareness between men and women in the US. The more recent time trends indicate that the previously described sex difference has been diminishing, along with an overall increase in CKD awareness. Our results are not necessarily in contrast with a recent report [24] which did not identify a significant rise in CKD awareness, as this previous study aggregated the NHANES cycles differently, thereby 'smoothing out' the 2015–2016 cycle and moreover did not include the 2017–2018 data, where we identified the strongest increase in awareness. The rise in CKD awareness levels in the US population was to a large part driven by growing CKD awareness rates in Caucasian women (from 8.5% in the first to 18.0% in the last 4-year period considered), while the improvement in the sex difference among African Americans was markedly smaller. Although this finding might be a consequence of differing education levels, African Americans had generally higher awareness rates than Caucasians, perhaps related to better communication of the risk associated with CKD within the African American community, which is however a purely speculative thought.

In agreement with numerous reports from diverse countries and population-based studies, more women than men have an eGFR indicative of CKD stages G3—G5 [10]. Yet, our findings update previous reports [22–24] and suggest that despite this increased disease prevalence among women compared with men, CKD awareness is consistently lower amongst women. Examining serum creatinine alone is an insensitive method of detecting CKD [34], and it could be postulated that the naturally lower creatinine in women (who tend to have lower muscle mass than men) has been interpreted as an absence of CKD. Equations have been developed that use serum creatinine in addition to [in the case of the nowadays most often used CKD-EPI formula [35]] age, sex and race to estimate the GFR, but not all physicians may apply these formulas at the bedside. Automatic eGFR reporting has been mandated progressively across most states since 2007 [36]. Although such reporting may increase the likelihood of false positive diagnoses [37], it may have contributed to fill this gap and explain the diminished gender gap over time in our study as well as the increased female awareness at lower serum creatinine levels (**Fig 4**). The diagnosis of CKD amongst the elderly is still a matter of debate, owing to the natural decline of kidney function with age, such that some authors propose that eGFR thresholds to define CKD should be different for old ages [12]. Owing to the longer life expectancy of women compared to men, we speculate that their low kidney function may be interpreted as "normal" for their age.

Apart from the sex-specific differences and the low CKD awareness overall, we were able to identify subgroups with a clear upward trend in CKD awareness over the last two decades. Awareness levels doubled in CKD stage G3, which may be an indicator for better early detection of CKD. Also for Non-Hispanic Whites, participants aged 80 years and older, participants without diabetes and subjects with hypertension, CKD awareness levels increased throughout the period of analysis. Most of the observed increases in CKD awareness within these subgroups, however, did not manifest until the last data-cycles (2015–2018). While in this univariate analysis the effects of the exposures intermingle, we think they are a sign of improvement.

Data from the United States Renal Data System (USRDS) show that women are as likely as men to receive nephrologist care before dialysis initiation, suggesting equivalent CKD awareness in men and women in late stage CKD, which is partially in contrast to the present findings. Specifically, in the 2019 USRDS Annual Data Report [38], the duration of pre-ESRD nephrology care (>12 months and other categories) was similar among men and women. However, these data are limited by the fact that they refer only to patients who ultimately start dialysis, and therefore do not contradict our conclusions.

## Limitations

Our study has several limitations. CKD is defined by abnormalities of kidney function, present for at least 3 months [39], but NHANES data are based on one-time measurements with no follow-up, which might lead to false positives. An overestimation of CKD due to misclassification, combined with a potentially low sensitivity of the CKD awareness question [40,41] would lead to an underestimation of CKD awareness. A previous study found that the CKD awareness question asked in NHANES has a sensitivity of only 33.2%, and that even a combination of five kidney-related questions only led to 61% sensitivity. However, the same study found 90.1% and 91.8% sensitivity for awareness of hypertension and diabetes, respectively, thus it is probably safe to assume that CKD awareness is still very low, when compared to other diseases.

Linked to the previous limitation is the fact that it is unclear what the CKD awareness question really measures, because CKD awareness is based on patient recall of what a doctor told them. The answer to the NHANES awareness question therefore depends in part on patient

recall and comprehension of the question and diagnosis, but also on healthcare access and testing, and communication of the results. As a related side note which might inspire further thinking, we were surprised by the finding that even among those with eGFR <15 mL/min/1.73m$^2$, not all individuals reported having been told they had weak or failing kidneys. (Due to the low number of participants with CKD stage G5, the volatility and decrease of awareness in both sexes in the last 4-year period analyzed [Table 1] should be considered with caution.) It would be interesting to find out if some patients, perhaps even under regular nephrologist care, might have answered "No" to the CKD awareness question (as some co-authors have previously experienced). However, as we mostly compare awareness between men and women within additional strata/groups, this limitation (not knowing what the CKD awareness question really measures) should affect men and women equally. In other words, when subgroups are compared or relative differences in CKD awareness are analyzed over time, these insights should be independent of an underestimated baseline.

As regards healthcare access as an explanatory factor for the present study findings, our analysis, regrettably, is limited by the fact that NHANES does not provide these data. Thus we were unable to assess the frequency by which women versus men visit primary care physicians or are referred for consultation to a nephrologist. Women might be referred to nephrologists less frequently than men, as men have been shown to predominate in several CKD cohorts, such as from the US [42], Canada [43], Japan [44], Germany [45], China [46] and France [47]. Less contact with nephrologists may therefore contribute to lesser awareness of CKD.

Another challenge lies in the NHANES data collection process. The 2-year NHANES data cycles are designed to be representative for the US population, based on a sample size of about ten thousand participants per cycle. Single individuals therefore carry large sample weights, which can be problematic when selecting only part of the data, such as CKD positive participants who are further dissected into subgroups. The finding that CKD awareness in CKD stage G5 dropped from 83.2% (men) and 69.2% (women) to 33.9 (men) and 25.3% (women) from the second last to the last period (see Table 1 and second paragraph of the Results section) seems unlikely and could be explained by the small sample size. The NHANES survey design has been found to oversample ethnic groups, low income groups and older people (in small variations, depending on the data-cycle) but does not oversample a certain sex, indicating that our main result of interest is likely correct [48].

## Conclusions

Despite the advances of the nephrology community in standardizing and simplifying the diagnosis and stratification of CKD [49], and after efforts in the dissemination of the importance of identification of the disease and understanding of its burdens to patients and healthcare systems [50], still only roughly 20% of CKD classified patients self reportedly claim to *ever have been told* they have weak or failing kidneys by a health professional, consistent with previous reports [22–24]. Due to the lack of signs and symptoms in the early stages of CKD, only active screening of CKD in high risk population will lead to early identification of the disease, which is currently recognized as cost effective in a high risk population [49]. Even after diagnosis, a mild reduction in eGFR may not be identified as a major problem, particularly in a population with multiple comorbidities. The worsening of kidney function leads to the appearance of clear signs and symptoms of CKD, and the lack of awareness in patients with eGFR <45 mL/min/1.73m$^2$, when referral to nephrologists in most regions becomes recommended, is more difficult to understand. The burden of CKD can hardly be overestimated, and progression of the disease can be slowed down for many individuals with guideline-based lifestyle and medical interventions [49]. Differences in CKD diagnosis and treatment between the sexes are

therefore not acceptable and should be tackled at every layer of healthcare services. As shown in this work, overall CKD awareness is growing, mainly because women are catching up to male awareness levels. This modest achievement to date needs to be accelerated further, such that a growing number of patients -men but especially women- learn of their disease at an early stage and can combat CKD in a timely manner.

## Supporting information

**S1 Table. Study sample characteristics of CKD positive and negative NHANES participants.** Study sample characteristics of CKD positive and CKD negative NHANES 1999 to 2018 participants who took part in the medical examinations; number of participants and weighted percentages.
(DOCX)

**S2 Table. Study sample characteristics of CKD stage G4 and G5 and CKD-negative NHANES participants.** Study sample characteristics of CKD stage G4 and G5 and CKD-negative NHANES 1999 to 2018 participants who took part in the medical examinations; number of participants and weighted percentages.
(DOCX)

**S3 Table. Adjusted CKD awareness, by patient characteristics and stratified by race/ethnicity.** This table refers to **Fig 2**, providing adjusted CKD awareness odds ratios (with 95% confidence intervals) for all (left), Caucasian (middle) and African American (right) study participants; odds ratios were adjusted for all other characteristics shown, odds ratios for all participants were further adjusted for race/ethnicity; within each model, women with one of the characteristics constituted the reference group.
(DOCX)

## Author Contributions

**Conceptualization:** Sebastian Hödlmoser, Wolfgang C. Winkelmayer, Jarcy Zee, Roberto Pecoits-Filho, Ronald L. Pisoni, Friedrich K. Port, Allison Ton, Juan Jesus Carrero, Eva S. Schernhammer, Manfred Hecking.

**Data curation:** Sebastian Hödlmoser.

**Formal analysis:** Sebastian Hödlmoser, Robin Ristl, Simon Krenn, Michał Lewandowski, Manfred Hecking.

**Funding acquisition:** Manfred Hecking.

**Investigation:** Sebastian Hödlmoser, Friedrich K. Port, Amelie Kurnikowski, Eva S. Schernhammer, Manfred Hecking.

**Methodology:** Sebastian Hödlmoser, Jarcy Zee, Robin Ristl, Simon Krenn, Michał Lewandowski, Manfred Hecking.

**Project administration:** Manfred Hecking.

**Resources:** Eva S. Schernhammer, Manfred Hecking.

**Software:** Sebastian Hödlmoser, Robin Ristl.

**Supervision:** Wolfgang C. Winkelmayer, Roberto Pecoits-Filho, Friedrich K. Port, Bruce M. Robinson, Allison Ton, Eva S. Schernhammer, Manfred Hecking.

**Validation:** Sebastian Hödlmoser, Wolfgang C. Winkelmayer, Jarcy Zee, Roberto Pecoits-Filho, Ronald L. Pisoni, Friedrich K. Port, Bruce M. Robinson, Simon Krenn, Allison Ton, Juan Jesus Carrero, Eva S. Schernhammer.

**Visualization:** Sebastian Hödlmoser, Amelie Kurnikowski, Manfred Hecking.

**Writing – original draft:** Sebastian Hödlmoser, Manfred Hecking.

**Writing – review & editing:** Sebastian Hödlmoser, Wolfgang C. Winkelmayer, Jarcy Zee, Roberto Pecoits-Filho, Ronald L. Pisoni, Friedrich K. Port, Bruce M. Robinson, Amelie Kurnikowski, Michał Lewandowski, Allison Ton, Juan Jesus Carrero, Eva S. Schernhammer, Manfred Hecking.

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
