## [Decision Letter · Decision Letter 0]

20 Oct 2020

PONE-D-20-30112

Sex Differences in Chronic Kidney Disease Awareness among US Adults, 1999 to 2018

PLOS ONE

Dear Dr. Schernhammer,

Thank you for submitting your manuscript to PLOS ONE. After careful consideration, we feel that it has merit but does not fully meet PLOS ONE’s publication criteria as it currently stands. Therefore, we invite you to submit a revised version of the manuscript that addresses the points raised during the review process.

We look forward to receiving your revised manuscript.

Kind regards,

Florian Kronenberg

Academic Editor

PLOS ONE

Journal Requirements:

"Initials of the authors who received each award: MH

Grant numbers awarded to each author: KL754-B

The full name of each funder: FWF, Austrian Science Fund

URL of each funder website: https://m.fwf.ac.at/en/

5. Please remove your figures from within your manuscript file, leaving only the individual TIFF/EPS image files, uploaded separately.  These will be automatically included in the reviewers’ PDF.

7. PLOS requires an ORCID iD for the corresponding author in Editorial Manager on papers submitted after December 6th, 2016. Please ensure that you have an ORCID iD and that it is validated in Editorial Manager. To do this, go to ‘Update my Information’ (in the upper left-hand corner of the main menu), and click on the Fetch/Validate link next to the ORCID field. This will take you to the ORCID site and allow you to create a new iD or authenticate a pre-existing iD in Editorial Manager. Please see the following video for instructions on linking an ORCID iD to your Editorial Manager account: https://www.youtube.com/watch?v=_xcclfuvtxQ

8. Please include your tables as part of your main manuscript and remove the individual files. ** Please note that supplementary tables (should remain/ be uploaded) as separate "supporting information" files **.

Reviewers' comments:

Reviewer's Responses to Questions

**Comments to the Author**

1. Is the manuscript technically sound, and do the data support the conclusions?

Reviewer #1: Yes

Reviewer #2: Yes

Reviewer #3: Partly

2. Has the statistical analysis been performed appropriately and rigorously? 

Reviewer #1: Yes

Reviewer #2: Yes

Reviewer #3: N/A

3. Have the authors made all data underlying the findings in their manuscript fully available?

Reviewer #1: Yes

Reviewer #2: Yes

Reviewer #3: Yes

4. Is the manuscript presented in an intelligible fashion and written in standard English?

Reviewer #1: Yes

Reviewer #2: Yes

Reviewer #3: Yes

5. Review Comments to the Author

Reviewer #1: The authors examine NHANES data to conclude that CKD awareness is lower in women than in men and that the gap between the sexes is narrowing due to greater reliance on eGFR, replacing serum creatinine, as a measure of renal function

Major

1. NHANES does not provide data regarding access to healthcare. There are no data to assess the frequency which women versus men visit PCPs or are referred for consultation to a nephrologist. Prior publications indicate that women are referred to nephrologists less frequently than men and data from the USRDS suggests that women undergo fewer serum creatinine measurements than men. Less contact with PCPs and nephrologists and fewer serum creatinine measurements may contribute to lesser awareness of CKD. The authors should include this limitation in their discussion.

Minor

1. The authors point to a large body of literature that indicates that CKD is less prevalent in men than in women. In this context, they should discuss limitations of these data including (1) differences in bias between the sexes in the GFR estimating equations and (2) controversy regarding the need for sex/age specific GFR cut offs and UACR cutoffs to define CKD.

2. In the Introduction section the authors point out that women start dialysis at a lower GFR than men. However they should also point out that this observation does not account for the difference in bias between the sexes in the GFR estimating equations which is equivalent to the difference in the GFR at the start of dialysis in men vs. women.

3. The authors indicate that serum creatinine was corrected and cite several references -- NHANES Data Documentation, Codebook, and Frequencies. Since standardization of creatinine assays is an important issue in evaluating their data and conclusions, the authors should describe in some detail how the correction was performed.

4. The authors may wish to speculate as to why it is that Caucasian women drive the increase in CKD awareness over time.

5. The authors may wish to point out the weakness of data from the USRDS that seem to contradict their conclusions. USRDS data show that women are as likely as men to receive pre-ESRD care from a nephrologist, strongly suggesting equivalent CKD awareness in men and women in late stage CKD. However, these data are seriously limited since they only apply to individuals who ultimately start dialysis

6. In general the paper is clear, well written and easy to understand. However, on several occasions the authors misuse idioms that obfuscate the meaning of sentences. For example: “consecutively declining” and “tilt towards”. (Unfortunately the pages are not numbered so I cannot identify these phrases by page and line number).

Reviewer #2: Hödlmoser et al. investigated the sex difference in the awareness of CKD using data from US NHANES. Overall, the awareness was low <22.5%. The authors observed higher awareness in men than in women, but the sex-difference became smaller overtime. The sex difference was more evident in whites than in blacks. The study question is important and the manuscript is well written overall. Nonetheless, I have several suggestions and comments as summarized below.

Major

1. The definition of CKD.

a. “CKD is characterized by a gradual loss of kidney function,” Kidney damage is a part of CKD, which may not be accompanied by reduced kidney function. This sentence thus should be rephrased.

b. “we restricted our study to CKD stages 3 to 5, which are based solely on eGFR” Please specify that these are KDIGO CKD G stages. This comment applies to the entire manuscript.

c.

2. Not sure whether the authors reported weighted results for the descriptive statistics in the section of “Characteristics of the study sample”. Please specify. If the authors reported unweighted estimates, they should report the weighted results instead.

3. Figure 2 was blank in the PDF.

4. Figure 2 may be trying to show this, but it seems important to show an analysis restricted to participants with diabetes, hypertension, or both.

Minor

Abstract

1. Please specify that sampling weights were taken into account in the Abstract.

Methods

2. “[20 to 50 years], [50 to 65 years], [65 to 80 years] and 80+ years” Not clear which category included 50, 65, and 80 years old.

Results

3. “most of the study participants (92.7%) had CKD stage 3” This seems misleading. I believe the authors meant “most of the study participants with CKD had G stage 3”

4. “Due to the low number of participants with CKD stage 5, the volatility and decrease of awareness in both sexes should be considered with caution.” This is not a result and thus should be moved to Discussion.

5. “Figure 4 s” s should not be in bold.

Discussion

6. The authors should discuss higher awareness in blacks than in whites.

7. “Chronic kidney disease is defined by abnormalities of kidney function,” The defined abbreviation of CKD should be used.

Reviewer #3: The review is too difficult to read. The main problem is due to a different diseases that can lead to CKD. For example the glomerulonephrithis should be considered. Also CKD should be diagnosed also considered urine output, proteinuria, ecc

6. PLOS authors have the option to publish the peer review history of their article (what does this mean?). If published, this will include your full peer review and any attached files.

Reviewer #1: No

Reviewer #2: No

Reviewer #3: No

---

## [Author Response · Author response to Decision Letter 0]

11 Nov 2020

Please find a detailed point-by-point rebuttal in our attachment entitled "Response to Reviewers". We could not paste and copy here because the letter comprises figures.

---

## [Decision Letter · Decision Letter 1]

23 Nov 2020

Sex Differences in Chronic Kidney Disease Awareness among US Adults, 1999 to 2018

PONE-D-20-30112R1

Dear Dr. Schernhammer,

We’re pleased to inform you that your manuscript has been judged scientifically suitable for publication and will be formally accepted for publication once it meets all outstanding technical requirements.

Kind regards,

Florian Kronenberg

Academic Editor

PLOS ONE

Additional Editor Comments (optional):

Reviewers' comments:

Reviewer's Responses to Questions

**Comments to the Author**

1. If the authors have adequately addressed your comments raised in a previous round of review and you feel that this manuscript is now acceptable for publication, you may indicate that here to bypass the “Comments to the Author” section, enter your conflict of interest statement in the “Confidential to Editor” section, and submit your "Accept" recommendation.

Reviewer #1: All comments have been addressed

Reviewer #2: All comments have been addressed

2. Is the manuscript technically sound, and do the data support the conclusions?

Reviewer #1: (No Response)

Reviewer #2: Yes

3. Has the statistical analysis been performed appropriately and rigorously? 

Reviewer #1: (No Response)

Reviewer #2: Yes

4. Have the authors made all data underlying the findings in their manuscript fully available?

Reviewer #1: (No Response)

Reviewer #2: Yes

5. Is the manuscript presented in an intelligible fashion and written in standard English?

Reviewer #1: (No Response)

Reviewer #2: Yes

6. Review Comments to the Author

Reviewer #1: (No Response)

Reviewer #2: Overall, the authors have adequately addressed my comments. I do not have any other additional comments.

7. PLOS authors have the option to publish the peer review history of their article (what does this mean?). If published, this will include your full peer review and any attached files.

Reviewer #1: No

Reviewer #2: No

---

## [Editor Report · Acceptance letter]

7 Dec 2020

PONE-D-20-30112R1 

Sex differences in chronic kidney disease awareness among US adults, 1999 to 2018 

Dear Dr. Schernhammer:

I'm pleased to inform you that your manuscript has been deemed suitable for publication in PLOS ONE. Congratulations! Your manuscript is now with our production department. 

Kind regards, 

on behalf of

Professor Florian Kronenberg 

Academic Editor

PLOS ONE